# Factors leading to disparity in lung cancer diagnosis among black/African American communities in the USA: a qualitative study

Nicholas Thuo,[1] Tanimola Martins ![ORCID] ,[2,2] Eugene Manley, Jr,[3] Maisha Standifer,[4] Dawood H Sultan,[5] Nicholas R Faris,[6] Angela Hill,[7] Matthew Thompson ![ORCID] ,[1] Rohan Jeremiah,[8] Morhaf Al Achkar ![ORCID] [9]

For numbered affiliations see end of article.

**Correspondence to**
Dr Morhaf Al Achkar;
alachkarm@karmanos.org

## ABSTRACT

**Objective** This study has two objectives: first, to explore the diagnostic experiences of black/African American (BAA) patients with lung cancer to pinpoint pitfalls, suboptimal experiences and instances of discrimination leading to disparities in outcomes compared with patients of other ethnic backgrounds, especially white patients. The second objective is to identify the underlying causes contributing to health disparities in the diagnosis of lung cancer among BAA patients.

**Methods** We employed a phenomenological research approach, guiding in-depth interviews with patients self-identifying as BAA diagnosed with lung cancer, as well as caregivers, healthcare professionals and community advocates knowledgeable about BAA experiences with lung cancer. We performed thematic analysis to identify experiences at patient, primary care and specialist levels. Contributing factors were identified using the National Institute of Minority Health and Health Disparities (NIMHD) health disparity model.

**Results** From March to November 2021, we conducted individual interviews with 19 participants, including 9 patients/caregivers and 10 providers/advocates. Participants reported recurring and increased pain before seeking treatment, treatment for non-cancer illnesses, delays in diagnostic tests and referrals, poor communication and bias when dealing with specialists and primary care providers. Factors contributing to suboptimal experiences included reluctance by insurers to cover costs, provider unwillingness to conduct comprehensive testing, provider bias in recommending treatment, high healthcare costs, and lack of healthcare facilities and qualified staff to provide necessary support. However, some participants reported positive experiences due to their insurance, availability of services and having an empowered support structure.

**Conclusions** BAA patients and caregivers encountered suboptimal experiences during their care. The NIMHD model is a useful framework to organise factors contributing to these experiences that may be leading to health disparities. Additional research is needed to fully capture the extent of these experiences and identify ways to improve BAA patient experiences in the lung cancer diagnosis pathway.

## STRENGTHS AND LIMITATIONS OF THIS STUDY

⇒ Strengths of this study included leveraging the perspectives of lung cancer survivors, caregivers and frontline healthcare providers to provide diverse, real-life insights.
⇒ Robust frameworks, including the cancer diagnosis pathway and National Institute of Minority Health and Health Disparities research framework, were used to organise the findings.
⇒ The voices of black/African American patients were prioritised to provide transparent, authentic perspectives.
⇒ Limitations included a highly educated, insured population and a small sample size, potentially missing the views of diverse and underserved patients.

## WHAT IS ALREADY KNOWN ON THIS TOPIC:

⇒ Black/African Americans (BAAs) have the highest morbidity and mortality rates for most cancers compared with any other racial group. Addressing the root causes of sub-optimal experiences for BAAs in cancer diagnosis pathways can help mitigate such disparities.

## WHAT THIS STUDY ADDS:

⇒ Our study delves into the experiences of patients with lung cancer, caregivers, and healthcare providers along the diagnostic and treatment pathways, highlighting the factors contributing to sub-optimal experiences.

## HOW THIS STUDY MIGHT AFFECT RESEARCH, PRACTICE, OR POLICY:

⇒ A deeper understanding of these experiences and the factors influencing them could lead to positive changes in provider-patient interactions, identify areas for improvement in provider training and health systems, and improve cancer diagnosis and treatment pathways for BAAs.

## INTRODUCTION

Lung cancer disproportionately affects black/African American (BAA) communities



in the USA.[1–5] In 2022, BAA had a lower estimated 5-year survival rate (20%) compared with whites (22%).[5] BAA in the USA have a lower rate of early-stage detection for lung cancer, at only 16% (compared with 20% among all races), with 53% being diagnosed at distant locations (compared with 49% for non-Hispanic whites).[5] BAAs are less likely than patients from other racial/ethnic groups to be staged during diagnosis, and to receive curative treatment at the most optimal point in the course of the disease.[2 6 7] These disparities appear to be unrelated to health insurance, as BAA Medicare beneficiaries with lung cancer experience higher mortality rates than non-Hispanic whites.[8]

The National Institute of Minority Health and Health Disparities (NIMHD) research framework offers guidance on considering the domains and levels of influence of factors that may account for why lung cancer disproportionately affects BAAs.[9] Differences in nicotine metabolism, which are linked to genetics and heredity, have been identified as a potential biological vulnerability.[1 10 11] Documented behavioural factors include smoking patterns, exposure to secondhand smoke and care avoidance due to beliefs and fears.[1 12] Environmental factors, including greater exposure to radon and indoor pollutants, living near smelting factories, and working in coal mines, have been identified as potential contributing factors to these disparities.[13] Lower educational attainment, rurality, poverty and limited access to healthcare due to employment are significant implicated sociocultural factors.[14–16] Research further indicates that systemic racism leading to inequitable access to quality healthcare is the critical factor for the gap in outcomes when a BAA develops lung cancer.[17 18]

Routine use of molecular testing and targeted cancer therapy for late-stage non-small cell lung cancer (NSCLC) has significantly reduced mortality rates.[19 20] Patients who receive molecular testing within 60 days of diagnosis have better survival rates compared with those who do not.[21] Studies have found that BAAs have similar rates of oncogenic alterations as non-Hispanic whites and at least one in every three BAA patients have an actionable mutation.[22] Nevertheless, BAAs tend to receive molecular testing or targeted treatment less frequently than whites.[21] Even in Medicare populations, compared with non-Hispanic whites, BAA patients over 65 years old with NSCLC had lower ORs of 0.53–0.63 for receiving targeted therapy.[23 24] White patients with NSCLC were more likely to receive next-generation sequencing, compared with BAA patients (50.1% vs 39.8%, p<0.0001).[25] Limited access to testing among BAA patients may be contributing to disparities in outcomes.[26] Studies have shown that when given access to care, BAA lung cancer patients have the same outcomes as white patients.[22 27]

Extensive literature exists on the roots of disparities and how to eliminate them for patients with early-stage lung cancers.[2 28–30] However, little is known about the aetiologies of disparity for patients with late-stage lung cancer. While it is possible that aetiologies of disparities in treating metastatic diseases share some commonalities with early-stage cancers, they may also have unique attributes contributing to the complex and long-term treatment paradigm of metastatic diseases. This study aimed to achieve two objectives: the first was to explore the diagnostic pathways for BAAs with lung cancer to identify pitfalls, suboptimal experiences and discriminatory practices occurred. The second objective was to identify the aetiologies of health disparity in cancer diagnosis for BAAs with lung cancer.

## METHODS
### Study setting, population and design
We used a phenomenological research design to guide qualitative interviews conducted with patients, caregivers, healthcare professionals and community advocates.[31] Patient and caregiver interviews provided patient-level accounts to explore experiences and identify diagnostic pathways and challenges encountered, from which we identified suboptimal experiences during the diagnostic phase. Healthcare professional and community advocate interviews produced reflections and observations to contextualise possible causes of observed challenges for individuals in BAA communities. The study was co-signed and conducted with the help of a community advisory group (CAG) called Project RADICAL—RAcial DIsparities in CAncer of the Lung—formed in 2020. The CAG, consisting of patients, caregivers, lung cancer researchers and community advocates, met biweekly to develop the study protocol, inform study methods and interview guides, and interpret findings.

### Patient and public involvement
We used a purposive sampling strategy to identify participants who met the inclusion criteria, which included patients aged 18 years or older who self-identified as BAA and had been diagnosed with metastatic lung cancer in the USA. Caregivers were individuals providing primary care for a BAA lung cancer patient. Healthcare providers (HCPs) and community advocates were practicing physicians providing oncology or primary care services, or advocates for BAA communities. To identify participants, we used snowball sampling methods through the community network of Project RADICAL CAG members, collaborators' networks in cancer organisations, online support groups such as ALK-positive, and academic list servers focused on disparities.

### Data collection
The research team developed the interview guides using Anderson's model of pathways to cancer treatment.[32] Based on the findings of our previous work,[31] the model was refined to organise patients' diagnostic experiences into three phases: (1) patient phase (from first symptom recognition to the first doctor's visit in primary care), (2) primary care phase (from the first primary care visit to presumptive diagnosis based on imaging) and (3)

| ID | Age group | Gender | Race/ethnicity | Education | Zipcode | Marital status | Income (000 $) | Role |
|---|---|---|---|---|---|---|---|---|
| 101 | 25–44 | Female | BAA | Doctorate | 11203 | Single | <50 | Patient |
| 102 | 45–64 | Female | BAA | Some college | 33705 | Married | 50–100 | Caregiver |
| 103 | 45–64 | Male | BAA | Doctorate | 46268 | Married | 100–150 | Researcher |
| 104 | >65 | Female | BAA | Doctorate | 60621 | Divorced | 50–100 | Patient* |
| 105 | >65 | Female | BAA | Masters | 60652 | Married | 50–100 | Caregiver |
| 106 | 25–44 | Female | BAA | Bachelors | 10701 | Single | 100–150 | Patient |
| 107 | 45–64 | Female | BAA | Bachelors | 46234 | Married | 100–150 | Patient |
| 108 | 25–44 | Female | BAA | Bachelors | 46268 | Married | 50–100 | Advocate |
| 109 | 45–64 | Male | BAA | Some college | 91367 | Single | 50–100 | Patient |
| 110 | 45–64 | Male | BAA | Masters | 46278 | Married | 100–150 | Caregiver |
| 111 | 45–64 | Female | BAA | Doctorate | 46214 | Married | >150 | Caregiver |
| 112 | Unknown | Male | BAA | Doctorate | Unknown | Married | >150 | Oncologist |
| 113 | 25–44 | Female | BAA | Doctorate | 31008 | Married | >150 | Provider |
| 114 | 45–64 | Female | BAA | Doctorate | 15146 | Married | >150 | Provider |
| 115 | 45–64 | Male | White | Doctorate | 43452 | Married | >150 | Provider |
| 116 | 45–64 | Male | White | Doctorate | 80111 | Married | >150 | Provider* |
| 117 | 45–64 | Male | White | Doctorate | 98115 | Married | >150 | Provider |
| 118 | 45–64 | Female | White | Master's | 41042 | Single | $50k–$100k | Nurse navigator |
| 119 | 45–64 | Female | BAA | Doctorate | 33579 | Married | >150 | Caregiver |

**Table 1** Participants' demographics

*Participants who were retired, all other participants were in full-time employment.
BAA, black/African American.

secondary care phase (from referrals to biopsy until identifying actionable genes and starting treatment).[33] The patient and caregiver interviews began with open-ended questions about their experiences with a cancer diagnosis. HCP and community advocate interviews aimed to identify factors contributing to suboptimal experiences along the diagnosis pathway (see online supplemental appendix 1 for interview guides). At the end of the interviews, participants shared their demographics, including the race they identify with. An experienced qualitative researcher (MA) conducted all interviews via phone or video conference, with most conducted via Zoom. Interviews were audio recorded and transcribed verbatim. Participants received a $50 gift card for participating in the 30–60 min interviews.

## Analysis

An experienced qualitative researcher (NT) coded the interviews using Dedoose (Dedoose Inc., Manhattan Beach, California) for analysis. The senior author (MA) provided peer debriefing and supervised NT's coding. Transcripts were analysed using inductive and deductive strategies, with codes developed highlighting suboptimal experiences and factors leading to disparity. The NIMHD research framework was used to organise the emerging factors contributing to disparity, and patient and caregiver narratives were triangulated with community advocates and HCPs' reflections. Findings were presented to the CAG and coauthors, and inputs were used to iteratively develop themes.

## Researchers' characteristics and reflexivity

The research team brings diverse backgrounds, experiences and perspectives to this qualitative study. MA is a male, PhD-trained qualitative researcher, identifying as an Arab immigrant, with personal experience living with advanced lung cancer. MS is a female, PhD-trained qualitative researcher, identifying as BAA, and has served as a caregiver for a BAA patient who succumbed to advanced lung cancer. NT, an MPH-trained and experienced qualitative researcher, identifies as Black African from Kenya. TM is a male, PhD-trained qualitative researcher, identifying as Black African from Nigeria. RJ is a male, PhD-trained researcher with expertise in qualitative research, identifying as BAA. DHS is a male, PhD-trained health service researcher, identifying as BAA. NF holds a Master of Divinity and serves as a research manager and health equity leader; he identifies as a white male. AH is a female, PharmD with experience in disparity research, identifying as BAA, and has had a family member who died of advanced lung cancer. EMJ is a male, PhD-trained molecular and cell biologist and researcher, identifying as BAA, and has studied lung cancer therapeutics and is a patient advocate. MT is a male, PhD-trained family doctor and researcher, who has also experienced the loss of a family member to lung cancer.

**Figure 1** Suboptimal experiences throughout the diagnosis pathway. PCP, primary care provider.

## RESULTS

The demographic characteristics of the participants are presented in table 1. Participants were aged 25–64 years old, with 55% female, 78% self-identified as BAA and 58% were patients or caregivers. At the time of the interview, 94% were in full-time employment. We will describe the themes that emerged in two main areas, namely suboptimal experiences and factors leading to disparities.

### Suboptimal experiences along the diagnosis pathways

We organised the suboptimal experiences within the patient phase, the PCP/hospital/emergency room (ER)/urgent care phase and the specialist phase. Figure 1 depicts suboptimal experiences throughout the diagnosis pathway. Table 2 includes supportive quotes.

### Patient phase

▶ *Not being alarmed by the symptoms as they appeared.* Some patients did not seek care right away as they were not concerned about the symptoms either because of their mild nature or the failure to appraise their seriousness.
▶ *Explaining symptoms as caused by other conditions.* Patients and caregivers at times attributed symptoms to less serious health conditions, such as common cold and allergies, delaying seeking medication consultation.
▶ *Prolonging self-managing of symptoms.* Patients resorted to home remedies or over-the-counter treatment for cancer-related symptoms (eg, cough and pain) instead of seeking medical care.
▶ *Seeking care only as symptoms became severe.* The threshold to seek medical advice was not crossed until symptoms were too serious to ignore or especially bothersome (eg, severe pain).

### PCP/hospital/ER/urgent care phase

▶ *Being treated for common, more benign conditions.* Patients frequently received treatment for more common and less serious conditions, such as sinus infections or acid reflux. Patients then ignored the symptoms without clear instructions or a follow-up plan.
▶ *Delayed or not being offered diagnostic imaging.* Some patients were not offered diagnostic tests due to the HCP's low levels of suspicion for lung cancer.
▶ *Not being informed of abnormal findings on tests.* Some patients were not informed about abnormal findings on their diagnostic tests and were left without any follow-up. On occasions, patients were not informed about the findings or the plan and they discovered them by reading the handouts.
▶ *Communicated information poorly and left with uncertainty.* Patients were kept uninformed regarding the plans of care. Sometimes they were not told how they could receive help if they needed urgent care or had questions.

### Secondary care phase

▶ *Difficulty finding an appointment to see the specialist.* Some patients, especially in underserved areas, had difficulty finding timely appointments to see specialists.
▶ *Poor delivery of information by the HCP.* Some patients were hassled during their care, receiving sub-par communication or little guidance to make decisions.
▶ *Not receiving molecular testing.* At times, patients were not offered molecular testing—a standard of care—and as a result, missed an opportunity to receive novel treatments.
▶ *Feeling steered to receive unnecessary chemotherapy.* Some patients felt they were pushed to take the only option of chemotherapy, while other more novel treatment options were not discussed.

**Table 2** Supportive quotes for the suboptimal experiences along the diagnosis pathways

| Patient phase | PCP/hospital/ER/urgent care phase | Specialist phase |
|---|---|---|
| **Not being alarmed by the symptoms as they appeared** | **Being treated for common, more benign conditions** | **Difficulty finding appointment to see the specialist** |
| Time went on and cough never got better, but it did not seem to get worse (Patient 107) | They'll say, 'Hey, it's sinus or allergies', they'll give them an antibiotic (Nurse navigator 118) | If you live in a rural community and you're the only doctor in that community and the nearest specialist is 2 hours or 45 minutes away that can have an effect on your referral process and your ability to refer at times (Provider 113) |
| | My primary care doctor diagnosed me with having allergies as well as acid reflux. I was treated with Protonix. I was treated with Claritin, over-the-counter stuff and eventually I would just ignore it (Patient 107) | |
| **Explaining symptoms as caused by other conditions** | **Delayed or not being offered diagnostic imaging** | **Poor delivery of information by the provider** |
| They had similar symptoms before and they resolved on their own without seeking help or with conservative measures; may be that if they have ignored in the past that can lead to worst consequence for them (Provider 106) | I went to another doctor and the first thing the doctor asked me 'had you had an x-ray?' I was like, 'No. Nobody sent me for an x-ray' (Patient 106) | It was just another bad experience where I was sitting in the doctors for my appointment. He was perpetually on the telephone talking to other people as I was sitting there and then kind of squeezing in like when he was on hold or between calls telling me something really quick (Patient 109) |
| She had a cough for quite a while and we thought that it was allergies. Then, unfortunately, she was in a head-on collision with a drunk driver and it broke the transverse process of her C7 vertebra (Caregiver 104) | You can't do chest CTs in every person, that's not a reasonable approach. But I think that's, in part, where some diagnostic delays happen is that perhaps they didn't have a high enough pretest probability to do the subsequent downstream test (Provider 117) | I would think knowing that I am also in a medical field that you would have at least tried to explain to me. I did not think that I should have had to be like, I'm not leaving until somebody explains what happens. I do not think that I should have had to do that (Patient 101) |
| **Prolonging self-managing of symptoms** | **Not being informed of abnormal findings on tests** | **Not receiving molecular testing** |
| He could not go to sleep at night so he would be taking NyQuil and it was more of a consistent thing (Caregiver 102) | When I looked at that hospital discharge summary and saw that there were lung nodules there. I had not seen that before or ever been told or followed up about that (Caregiver 102) | I had one or two patients diagnosed with lung cancer who went to a smaller local facility for their follow-up, and I brought them back into the office to go through things. And I said, 'Well, I don't see any molecular testing on your biopsy'. You've lost that opportunity (Thoracic oncologist 115) |
| **Seeking care only as symptoms became severe** | **Communicated information poorly and left with uncertainty** | **Feeling steered to receive unnecessary chemotherapy** |
| One night I woke up because I felt like my left side of the chest was closing down on me, I felt that my left side was not getting enough air in (Patient 101) | Nobody talked to her about the mass, but she was reading her orders and she was like, 'What is this in my orders?' (Caregiver 111) | They were very pushy with the chemotherapy which made me even more skeptical. I was just saying, 'Wait a minute. I would like a second opinion'. I had texted the doctor and said, 'they want to do chemo immediately'. She said, 'No, let us wait. I will get him in next week and we can do a genetic test that can see if chemo is the best for him or targeted medicine'. That was the first time I had heard about targeted medicine (Caregiver 102) |
| | So who do I call? Which doctor's office do I call? If I'm having a problem, do I go to the ER? Do I go to Urgent Care? Do I go to the pulmonologist because my cousin goes to one? Or do I go to my primary care doctor? (Provider 113) | |

ER, emergency room; PCP, primary care provider.

## Factors leading to disparity in lung cancer diagnosis

Here, we describe the factors that lead to suboptimal experiences along the pathways to diagnosis and treatment for BAA patients. These factors work at multiple individual, interpersonal, community and societal levels. Table 3 includes supportive quotes.

### Individual factors

► *Not seeking timely care.* Some patients reported an aversion to seeking timely medical care for fear of finding serious illnesses or having to deal with adverse outcomes.

► *Inadequate reporting of relevant symptoms.* The patients' experiences with disease symptoms influence their decisions on which symptoms to report. They may not feel comfortable reporting certain symptoms and may understate the details.

► *Logistic constraints to accessing healthcare.* Patients living in poorer areas tend to have poorer healthcare outcomes due to affordability and income issues. At

**Table 3** Supportive quotes for the factors leading to disparity in lung cancer diagnosis

| Individual factors | Interpersonal factors | Community factors | Societal factors |
|---|---|---|---|
| **Not seeking timely care** | **Lack of compassionate family support** | **Cultural practices concerning healthcare** | **Working in lower-paying jobs** |
| Yeah. I mean, I think it's related to, 'I hope that things will get better!' It can also be related to a form of denial So, in fact, they know it might be lung cancer and are not going in as a form of denial (Provider 117) | In the majority in a community, you're more likely to have a wider network within that community. If you have a cough, you're more likely to be talking to somebody about it and somebody said, 'You should get it checked out'. If you are more isolated within the community because of race or economic issues(you won't have)that network (Provider 115) | Some people take their symptoms more seriously than others. I don't know whether this is more or less among African Americans, but to the extent to which symptoms are minimized or ignored, that's more personal or cultural (Provider 117) | A lot in African-American and Hispanic Communities do not have insurance, rather because they cannot afford healthcare insurance. Because they are making minimum wage jobs or it just not on the priority list of 'I have never been sick (Advocate 108) |
| **Inadequate reporting of relevant symptoms** | **Doctors' schedules not accommodating patients** | **Availability of clinical services** | **Cost of receiving care** |
| What's important to the patient and to the doctor are two different things. So, they may not consider their shortness of breath and issue because they've been having it over a period of time (Provider 113) | A lot of jobs, you know, the timing, like the doctor's offices, closes at five o'clock, and they normally take their last patient at 4.30 (Advocate 108) | I think it's also where is the setting of the low-dose CT scans. Most of them are in hospitals, or some patients just won't go to a hospital for care (Provider 114) | They try to wait it out because they have a job, they can't take off work without taking PTL, or just they're doing something else for their family (Nurse navigator 118) |
| **Logistic constraints to accessing healthcare** | **Societal pressure to seek non-medical approaches** | **Disjointed clinical services** | **Limited or variable insurance coverage** |
| If I want to get somebody screened for lung cancer, they may have to travel to a hospital 30 minutes away and not have even the gas money. If I want to get somebody for a biopsy and for specialty care, they're potentially looking at traveling an hour (Provider 115) | Sometimes patients will say, 'well you know, God heals and God will heal me' or they'll say 'I want to try an herbal natural remedy' (Provider 113) | Then with a diagnosis just sending her home with that order, I think was irresponsible. Somebody should have said, 'Hey, do you know you have this on your lung', and nobody did (Caregiver 111) | Well, this is really being driven by the insurance company. Your doctor really making some of these decisions. The insurance company made it for them. And that doesn't fit with your 'medical condition' at the moment (Caregiver 110) |
| **Avoiding medical care due to a previous encounter involving racial discrimination** | **The patient's surroundings normalise tolerating symptoms** | **Delays in obtaining appointments** | **Policies and laws determining investment in healthcare** |
| People have shared that they felt their expression of their pain level wasn't taken as seriously, because there might be some unconscious biases on the part of the healthcare provider about what they think about Black people and their threshold for pain tolerance (109) | I think sometimes people don't think things are serious because it's all perception. So, if you have a family member, who let's say, everybody in your household smokes and everybody complains of shortness of breath, you probably think that it's normal to be short of breath (113) | Just figuring that out the next step of scheduling an appointment having to get on the phone and wait to schedule an appointment. It's not as easy as getting on to your computer sometimes, depending on the Health Care system (113) | It relates to what resources are available in your area. If you live in a rural community and you're the only doctor in that community and the nearest specialist is 2 hours away that can have an effect on your referral process (113) |
| **Cultural identity** | **HCPs indulging in discriminatory practices** | **Organisational demands on HCPs to see more patients** | **Limited management using telehealth** |
| Sometimes it is just simply cultural beliefs. They don't want to admit that they have a problem. I mean, that could be something as simple as they have a pain somewhere. You know, i'll just take Tylenol or I'm coughing I won't go to the doctor, or they'll say things like you know (Provider 113) | So, at the doctor level and, across cultures, they see an African-American person and they may or may not be obese, but there's so much emphasis on 'You need to lose weight. You need to exercise. You need to eat right'. And again, might be not investigating the extent by which the drinking and the smoking (Caregiver 119) | If you have a new patient and someone telling you have to see patients every 15 minutes and you're at the end of the day or the middle of day and you're just trying to get through it. The provider can feel rushed and glazed over the patient's symptoms (Provider 113) | He ended up going to get some antibiotics via telemedicine from a doctor. But I think I noticed more more persistent coughing (Caregiver 102) |

**Table 3** Continued

| Individual factors | Interpersonal factors | Community factors | Societal factors |
|---|---|---|---|
| **Lack of health literacy** | **Patients anticipating prejudice and discrimination** | | **Practitioner's inexperience** |
| If there's no understanding of medicine, medical illness would be deemed as something annoying, like a cough. Not understanding risk factors for disease. Based on either smoking or family history or other people with those symptoms, not understanding when to seek care (Provider 114) | It was awful. If he had taken the opportunity to send me for an x-ray and I had great insurance. It's not like I was like, hey, let's penny-pinch. I do not want to do this, or I do not want to do that'. I followed all of the instructions that he had given me, down to the tee and it was never an option for him to send me for an x-ray (Patients 106) | | I truly believe that if you recognize something you order it in a timely manner, but if you don't recognize it, I think part of it is recognition of the problem if you don't recognize it, it can delay you (Provider 113) |
| **Not knowing which HCP to contact** | | | **Complex health systems not designed for vulnerable patients** |
| Finding a doctor is so messed up. You get no information on a doctor. You are given a list of names and you are supposed to pick one. Well, how am I supposed to pick them based on the name? That is the first barrier (Researcher 103) | | | You have been in a hospital before. It is almost impossible to navigate a hospital because parking is difficult. Getting in the maze. Knowing where to go and where you cannot go is difficult (Researcher 103) |

HCP, healthcare provider.

times, patients live far from the health facility and are unable to access transportation or assistance to travel for treatment.

► *Avoiding medical care due to a previous encounter involving racial discrimination.* Perceptions of the HCP based on previous experiences often lead to mistrust and hesitancy among patients to avoid possible dismissals or maltreatment.

► *Cultural identity.* People are often influenced by their community's behaviour and religious beliefs regarding healthcare.

► *Lack of health literacy.* At times, patients are unable to appreciate the severity of a diagnosis or are unaware of what to do when they experience health changes.

► *Not knowing which HCP to contact.* Information on HCPs and their specialties are not readily available, making it hard for the patient to determine who to reach out to and where.

**Interpersonal factors**

► *Lack of compassionate family support.* For some individuals, weaker family ties lead to socioeconomic isolation, which obscures the management of their healthcare needs.

► *Doctors' schedules not accommodating patients.* In certain areas, the doctors' offices or health facilities are only open during weekdays, when many people are at work and are therefore unable to access healthcare.

► *Societal pressure to seek non-medical approaches.* It is often observed that the patient's social networks encourage the use of alternative treatment methods, including herbal and non-regulated remedies. As a result, some patients may choose spiritual practices or herbs over medical care.

► *Patients' surroundings normalise tolerating symptoms.* Patients seldom internalise the need to endure the disease and fear the loss of dignity through their dependence on others, especially where the symptoms are common around them.

► *HCPs indulging in discriminatory practices.* It has been observed that HCPs tend to neglect, miss, or dismiss symptoms reported by BAA patients, especially pain. Furthermore, they may not execute appropriate referrals.

► *Patients anticipating prejudice and discrimination.* Patients are concerned about how HCPs may treat them based on their personal attributes, especially race. Others, especially individuals who used to smoke, avoid seeking medical care due to the fear of being blamed for their illness and the associated stigma of causing cancer.

**Community factors**

► *Cultural practices concerning healthcare.* Certain therapies may be preferred in a community, for example, obtaining painkillers instead of identifying the cause of the ailment.

► *Availability of clinical services.* There is a dearth of hospitals, diagnostic facilities, primary care providers (PCPs), and specialists within certain communities and areas. This is especially true for smaller communities living in poor areas.

► *Disjointed clinical services.* Referrals and visits to different departments and hospitals may force patients to drop out of treatment and follow-up on results. Furthermore, communicating the results to PCPs after visiting the ER is left to the patient and not communicated directly, which often leads to missed opportunities for follow-up on diagnosis and treatment.

► *Delays in obtaining appointments.* Although booking systems for hospitals are now online, navigating these systems is complex and coupled with organisational constraints in the availability of appropriate staff. A patient's diagnosis may be delayed due to a lack of appointment slots.

► *Organisational demands on HCPs to see more patients.* For profitable practice, some hospitals and clinics set patient and charge goals for HCPs, which leads to poor service quality. Also, forcing HCPs to attend to more patients within a stipulated time may overwhelm them, reducing their quality of care.

### Societal factors

► *Working in lower-paying jobs.* The relegation of certain communities to low-paying jobs limits their access to resources for managing health and disease impacts. This economic disempowerment leaves personal healthcare needs competing for priority with one's family's needs. Often, family commitments deny the patient the appropriate time to seek treatment.

► *Cost of receiving care.* Seeking diagnosis and treatment is costly for individuals, especially when they face job insecurity and can take limited time off from work. This issue, coupled with the inflated costs of medical care, makes healthcare services inaccessible or undesirable.

► *Limited or variable insurance coverage.* Limited options within Medicaid coverage, cost of insurance, and the inability to obtain insurance led to suboptimal access to timely diagnosis and treatment. Besides, the unwillingness of insurance companies to pay for certain services, either in full or part, may limit one's options for quality care.

► *Policies and laws determining investment in healthcare.* The existing policies and laws related to healthcare and resources available to communities affect access, education and living conditions for the community.

► *Limited management using telehealth.* Only using telehealth and the absence of physical testing and comprehensive workups lead to missed diagnoses.

► *Practitioner's inexperience.* Underprivileged communities are often subjected to care by undertrained practitioners and a shortage of specialists. A lack of adequate skill and experience to manage chronic diseases may lead to lower suspicion or missing critical symptoms.

► *Complex health systems not designed for vulnerable patients.* Moving through complex online hospital systems and departments to book and access essential services requires internet access either by computer or phone, which can be challenging for many.

## DISCUSSION

This study is one of the first to report on the suboptimal experiences of BAA patients along the lung cancer care pathway. These experiences were identified by BAA patients, caregivers, and HCPs. The study found that some patients delayed seeking care, while others had their concerns ignored or were not offered molecular testing despite qualifying for it. Factors contributing to these suboptimal experiences included difficulties with insurance coverage, provider unwillingness to conduct comprehensive testing, provider bias in recommending treatment, high healthcare costs, and a lack of healthcare facilities and qualified staff to provide necessary tests and support. The study captured firsthand the individual and caregiver experiences, which were corroborated by community advocates and some HCP experiences, particularly those with the lived experience of being BAAs.

Our findings indicate that at least some of the patients in the study, all BAA, experienced delays in both the diagnosis and initiation of treatment. Some patients received treatment for other common diseases before tests revealed their lung cancer. These delays are consistent with the 2021 State of Lung Cancer report, which found that BAAs patients were 16% less likely to receive an early diagnosis and 7% more likely to go untreated compared with their white American counterparts.[34] HCPs' actions in offering diagnostic testing play a critical role in patient experiences. Positive relationships with HCPs can motivate patients to seek care, but we found that delays were sometimes caused by patients' concerns about discrimination from providers who may not refer them for timely diagnosis and targeted treatment. The literature shows that BAA patients often perceive communication with physicians as less supportive, less collaborative and less informative.[35] Our findings on communication and information sharing align with this, as we observed instances of critical information being withheld, findings being communicated in ways that were difficult for patients to understand, and information on available treatment options not being shared.

The factors we identified as contributing to delays in lung cancer diagnosis may also lead to poorer outcomes throughout the cancer care process.[36] BAA patients presenting at later stages of illness may experience delays for several reasons, such as limited health literacy and self-assessment of symptoms. Previous experiences with discrimination may also impact patients' decisions about seeking care. Additionally, patients have reported feeling that their providers did not consider their symptoms to be serious, resulting in the treatment of other possible infections instead of additional testing.[37 38] On the other hand, patients in our study reported using various

approaches to overcome the barriers they faced in their healthcare journey, including using their knowledge of health systems, seeking information on diagnostic and treatment options, and engaging with their insurance companies and HCPs to access comprehensive tests and treatment options. This underscores the importance of patient empowerment in advocating for their rights with insurance companies and HCPs, as well as the significance of receiving sufficient information from HCPs to make informed decisions about their healthcare. Patient navigators may also assist patients and families in navigating the health system and overcoming socioeconomic challenges. Addressing racism and prejudices through multi-level strategies, including retraining doctors, engaging policymakers, and educating non-clinical staff on prejudice, systemic biases and racism, is crucial.

This project helped mapping the range of suboptimal experiences, which can then be included in future surveys to screen patients for these occurrences. Future research should aim to collect experiences of disparity from a more representative population and develop community-led interventions targeting individual and systemic factors perpetuating disparity. Evidence-based research methods should also focus on collecting patient perspectives relevant to experiences of racism, prejudice and perceived injustices.

## CONCLUSION

This study revealed suboptimal experiences and factors contributing to the disparities in lung cancer care among BAAs. Delays in diagnosis and initiation of treatment were reported, and patients faced various barriers, including insurance coverage, provider bias and lack of healthcare facilities. BAA patients' empowerment and engagement with HCPs and insurance companies were important in navigating these barriers. Multilevel strategies are needed to address racism and prejudices in the healthcare system. Future research should focus on developing culturally tailored community-centred interventions and evidence-based initiatives to collect, analyse, and disseminate patient perspectives on experiences of racism and perceived injustices. Although there were limitations to the study, the triangulation of perspectives and the use of robust frameworks strengthened the findings.

**Author affiliations**
[1]Family Medicine, Univeristy of Washington, Seattle, WA, USA
[2]Health and Community Science, University of Exeter, Exeter, UK
[3]SCHEQ Foundation, New York, New York, USA
[4]Health Policy, Morehouse School of Medicine, Atlanta, Georgia, USA
[5]Public Health, Mercer University, Atlanta, Georgia, USA
[6]Multidisciplinary Thoracic Oncology Program, Baptist Cancer Center, Memphis, Tennessee, USA
[7]Pharmacotheraputics and Clinical Research, University of South Florida, Tampa, Florida, USA
[8]Global Health, University of Illinois at Chicago, Chicago, Illinois, USA
[9]Oncology, Wayne State University/Karmanos Cancer Institute, Detroit, MI, USA

**Contributors** The listed authors, NT, TM, EMJ, MS, DHS, NF, AH, MT, RJ and MA contributed to the conception, design, acquisition of data, or analysis and interpretation of data, drafting the article or critically revising it for important intellectual content, and final approval of the published version. NT and MA conceptualised this article and wrote the first draft along. NT, MA, TM, EMJ, NF, RJ, MS, DHS, AH and MT contributed to the revision and additional changes for the article to accurately reflect the research findings. MA affirms that the manuscript is an accurate representation of the issues covered, that there are no significant omissions or discrepancies. All authors have full responsibility for the work and decision to publish it.

**Funding** This research is linked to the CanTest Collaborative, which is funded by Cancer Research UK (C8640/A23385). The project was also partially funded by a gift from LUNGevity Foundation.

**Competing interests** None declared.

**Patient and public involvement** Patients and/or the public were involved in the design, or conduct, or reporting, or dissemination plans of this research. Refer to the Methods section for further details.

**Patient consent for publication** Not applicable.

**Ethics approval** This study involves human participants and ethics approval was obtained from the University of Washington (UW) Human Subjects Division Reference number STUDY00005438. Participants gave informed consent to participate in the study before taking part.

**Provenance and peer review** Not commissioned; externally peer reviewed.

**Data availability statement** Data are available upon reasonable request. The authors confirm that the data supporting the findings of this study are available within the article and its supplementary materials. The data will become available immediately after publication, with no end date for anyone who wishes access.

**ORCID iDs**
Tanimola Martins http://orcid.org/0000-0001-5226-4073
Matthew Thompson http://orcid.org/0000-0003-0256-8444
Morhaf Al Achkar http://orcid.org/0000-0002-4160-0550

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
