## [Reviewer comments · BMJ Open]

ARTICLE DETAILS

TITLE (PROVISIONAL)	Factors Leading To Disparity In Lung Cancer Diagnosis Among Black/ African American Communities in the United States: A Qualitative Study
AUTHORS	Thuo, Nicholas; Martins, Tanimola; Manley Jr., Eugene; Standifer, Maisha; Sultan, Dawood; Faris, Nick; Hill, Angela; Thompson, Matthew; Jeremiah, Rohan; Al Achkar, Morhaf

VERSION 1 – REVIEW

REVIEWER	Borondy Kitts, Andrea Rescue Lung Society
REVIEW RETURNED	17-Apr-2023

GENERAL COMMENTS	Factors Leading To Disparity In Lung Cancer Diagnosis Among Black/ African American Communities in the United States: A Qualitative Study Andrea Borondy Kitts MS, MPH 1. Is the research question or study objective clearly defined? “This study aimed to achieve two objectives: the first was to explore the pitfalls, suboptimal experiences, and discriminatory practices in diagnostic pathways for BAAs with lung cancer. The second objective was to identify the etiologies of health disparity in cancer diagnosis for BAAs with lung cancer.” The authors assume that there are pitfalls, suboptimal experiences, and discriminatory practices in the diagnostic pathways. I’m concerned that assuming there are suboptimal differences and discriminatory practices may have biased the researcher perceptions and analysis. It may also have biased the answers given by the study participants. It’s not clear if the study aims were discussed with the participants prior to their being interviewed. This needs to be addressed in the manuscript. It would have been preferable for the aims to “explore the experiences in diagnostic pathways for BAA’s with lung cancer.” This would have allowed for a less biased approach. 2. Is the abstract accurate, balanced and complete? In the methods section of the abstract the sentence: “Thematic analysis identified suboptimal experiences at patient, primary care, and specialist levels, along with contributing factors according to the National Institute of Minority Health and Health Disparities (NIMHD) health disparity model “ reads more like results. Recommend rephrasing “Used thematic analysis to identify experiences at patient, primary care and specialist levels.
---

	Contributing factors identified using the National Institute of Minority Health and Health Disparities (NIHMD) health disparity model.” 3. Is the study design appropriate to answer the research question? In a phenomenological study design the researchers is asked “to set aside their prejudices and a priori assumptions and focus mainly on the immediate experience.” The first aim of the study was to “explore the pitfalls, suboptimal experiences and discriminatory practices in diagnostic pathways for BAA’s with lung cancer.” The authors assume the participants experienced pit falls, suboptimal experiences and discriminatory practices. This seems to be biased and not in concordance with phenomenological study design practice. A more appropriate aim may have been to “explore the lived experience of BAA;s with lung cancer.” Please clarify. 4. Are the methods described sufficiently to allow the study to be repeated? Yes In the methods section – please delete the sentence “We identified suboptimal experiences during the diagnostic phase.” It belongs in the results. Page 6 line 54 please delete suboptimal as a descriptor for experiences. The descriptor assumes all experiences will be suboptimal. It would be helpful for the authors to describe the methods in objective terms. Page 7 line 25 – please reword “patient who succumbed to advanced lung cancer.” Infers that it was the patient’s fault that he/she died. Suggest rewording to patient who died of advanced lung cancer – same comment for line 33 5. Are research ethics (e.g. participant consent, ethics approval) addressed appropriately? Yes 6. Are the outcomes clearly defined? No – the outcomes do not describe how many study participants described experiences for each of the pathways 7. If statistics are used are they appropriate and described fully? Not applicable 8. Are the references up-to-date and appropriate? Suggest finding more recent publication. Bach PB, Cramer LD, Warren JL, Begg CB. Racial differences in the treatment of early-stage lung cancer. New England Journal of Medicine. 1999;341(16):1198-205 9. Do the results address the research question or objective? Yes 10. Are they presented clearly? Yes 11. Are the discussion and conclusions justified by the results No – the statement “Our findings indicate that the majority of patients experienced delays in both the diagnosis and initiation of treatment” is not supported as the authors don’t provide any indication of the number of participants that experienced any of suboptimal experiences or factors in the study design. There are 1 or 2 supportive quotes for each suboptimal experience category included and no quantitative information on how many of the
--	---

	participants had a suboptimal experience in each of the suboptimal experience categories. It would be helpful if the authors provided a count of how many patients/caregivers made comments about each of the suboptimal experiences along the diagnostic pathway and for each of the factors. With only one or two comments per experience and one per factor it's hard to assess how prevalent these issues were in the study. 12. Are the study limitations discussed adequately? No Included are small sample and highly educated and insured participants. Not included – potential for bias due to study aim assumptions that participants have had suboptimal experiences. See previous comments. 13. Is the supplementary reporting complete (e.g. trial registration; funding details; CONSORT, STROBE or PRISMA checklist)? Yes 14. To the best of your knowledge is the paper free from concerns over publication ethics (e.g. plagiarism, redundant publication, undeclared conflicts of interest)? Yes 15. Is the standard of written English acceptable for publication? Yes * Statistical review * 1. Does this paper require specialist statistical review? No Additional comments: Introduction: The sentence Whites, BAA patients over 65 years old with NSCLC had lower odds ratios of 0.53-0.63 for receiving targeted therapy “ seems to be out of context here. Please include references for the statement “Extensive literature exists on the roots of disparities and how to eliminate them for patients with early-stage lung cancers.” Table 1 – participant characteristics – highly educated – 12 of 19 have doctorate, 2 Masters, 3 Bachelors, and 2 with some college – how generalizable are the results? No discussion of smoking history – individuals who smoke or used to smoke are sometimes reluctant to seek care due to stigma of smoking and also more afraid of finding out they may have lung cancer. Suggest authors include smoking history in Table 1 and comment if participants who used to smoke or currently smoke expressed any self, implicit or explicit stigma related to their smoking history. Page 13 – individual factors “Logistic constraints to accessing healthcare. Patients living in poorer areas tend to have poorer healthcare outcomes due to affordability and income issues. At times, patients live far from the health facility and are unable to access transportation or assistance to travel for treatment.” Most of the participants had incomes >100k – 9 of 19 >150, 3 100-150k – was it an issue for only a few participants? Page 13 line 52 “It has been observed that HCPs tend to neglect,
--	--

	miss, or dismiss symptoms reported by BAA patients, especially pain. Furthermore, they may not execute appropriate referrals” Is this from participant comments or from the literature? Please clarify Page 15 line 6 – please use people first language – individual who smokes or smoked. Also is this point based on participant comments or on the literature. In the discussion of factors are these based on the literature or on lived experience from the healthcare professionals? Given the high education and socioeconomic status of the majority of the participants, many of these factors seem to not apply to this specific sample. It would be helpful to have demographic information on where the healthcare professionals practice (rural, urban, FQHC, academic medical center...) and more descriptive demographics on where the patients and caregiver participants live ie urban, rural, underserved not just zip code. Page 31 – typo in second paragraph “We want you help in ...” should be “We want your help in ...” There seems to be a typo on page 22 line 11 – the word “along” in the first sentence seems to be out of place. Page 21 line 9 – please replace “hard to reach patients” with “underserved patients.” The term “hard-to-reach” implies patient is at fault for not being easy to reach. Key Messages section: The key message on how this study may affect research, policy and practice seems to be an overreach. It is unlikely that a small study with 19 highly educated participants is generalizable enough to lead to “positive changes in provider-patient interactions, identify areas for improvement in provider training and health systems, and improve cancer diagnosis and treatment pathways for African Americans.” This study is more in the realm of hypothesis generation for future research that may lead to the improvements mentioned in the key message.
--	--

REVIEWER	Giaquinto, Angela American Cancer Society
REVIEW RETURNED	28-Apr-2023

GENERAL COMMENTS	This study provided qualitative interviews and evidence for adverse experiences among African American/Black people during the cancer care continuum specific to lung cancer. Overall, the study is well designed with clear and concise explanation of methods used. This reviewer suggests some additional information to be added to the results if collected to provide further evidence for statements given (see results section below). This reviewer also encourages the authors to review their manuscript for consistency in capitalizations. For instance, the introduction had variations in capitalizing “white”, Table 1 has inconsistent capitalizations of gender, race/ethnicity, employment, and role, etc. Table 2 had supportive quotes capitalized in the second and third column but not the first, etc. Below are additional comments for the authors’ consideration. Introduction: The introduction included a good background. Page 5 line 7-8, I encourage the researchers to use the most recent 5-year data (2016-2020) available from NCHS for mortality, or list the years that the given statistic is for as 55.4 and 46.9 per 100,000 for lung cancer death rates are not from 2016-2020. Page 5 line 27-30 has a sentence repeated, the first time has reference 16 and the duplicate sentence has references 17 and 18. Please double check references as I believe 16 should be referenced in the prior sentence beginning on line 24. Page 6 first paragraph, I encourage
---

	the authors in line 4 to include “diagnostic pathways for BAAs with metastatic/late-stage lung cancer.” To tie in the last paragraph on page 5. Methods: The methods were well written, clear, and concise. Results: If possible, this reviewer suggests that the authors include a table that gives qualitative evidence for claims, i.e. how many patients interviewed expressed similar individual factors that led to suboptimal experiences. Discussion states on page 19 line 18 that a “majority of patients experienced delays in diagnosis and initiation of treatment”. Can this data be included in the results? I.e., including a table or summarized findings from Question b viii. Discussion: Well written. This reviewer was surprised that 14 of the 19 interviewees had post-graduate education and only 1 patient reported an income less than 50,000-100,000. This highly educated and compensated population may not be generalizable, which the authors including in their limitations. This reviewer agrees that they study’s strength was the in-depth interviews and authentic perspectives. Overall, this is a well-written article that has a clearly defined study objective. I encourage the authors to include, if possible, more information on aggregated responses collected to further strengthen their discussion, i.e., how many patients had similar responses or experienced delays in treatment. This reviewer recommends the paper be accepted with minor revisions.
--	---

VERSION 1 – AUTHOR RESPONSE

Andrea Borondy Kitts MS, MPH

1. Is the research question or study objective clearly defined?

“This study aimed to achieve two objectives: the first was to explore the pitfalls, suboptimal experiences, and discriminatory practices in diagnostic pathways for BAAs with lung cancer. The second objective was to identify the etiologies of health disparity in cancer diagnosis for BAAs with lung cancer.”

The authors assume that there are pitfalls, suboptimal experiences, and discriminatory practices in the diagnostic pathways. I’m concerned that assuming there are suboptimal differences and discriminatory practices may have biased the researcher perceptions and analysis. It may also have biased the answers given by the study participants. It’s not clear if the study aims were discussed with the participants prior to their being interviewed. This needs to be addressed in the manuscript. It would have been preferable for the aims to “explore the experiences in diagnostic pathways for BAAs with lung cancer.” This would have allowed for a less biased approach.

Response: We appreciate the perspective of the reviewer. Indeed, we did anticipate that pitfalls, suboptimal experiences, and discriminatory practices existed based on the extensive literature concerning the experiences of African American lung cancer patients. Hence, this anticipation is not without foundation. However, our clarified objective was not to determine the existence of these challenges but to explore and characterize them. We did so by following the patients' experiences along their cancer diagnosis pathways. This approach is not equivalent to biasing the position.

The interview guide (please see attached) indicates that we openly asked about the experiences along the diagnosis pathway, walking step by step with the patients without

guiding their responses. Only towards the end did we ask patients to reflect on the experiences they had mentioned and specifically inquire if they had encountered pitfalls, suboptimal experiences, or discriminatory practices.

It is important to note that not every participant identified challenges at every step. The goal of this paper, however, is not to enumerate good practices. Rather, we aimed to identify and highlight what was suboptimal and needs improvement.

We acknowledge the possibility, albeit small, that our objective may have influenced some participants' focus on suboptimal experiences. In the interest of transparency, we communicated our study's objective to all participants.

This project's goal is not to confirm or disprove a hypothesis about whether these suboptimal experiences occur, or to what extent. These questions are for other studies to explore. This project aims to map the range of suboptimal experiences, which can then be included in future surveys to screen patients for these occurrences.

2. Is the abstract accurate, balanced and complete?

In the methods section of the abstract the sentence: "Thematic analysis identified suboptimal experiences at patient, primary care, and specialist levels, along with contributing factors according to the National Institute of Minority Health and Health Disparities (NIMHD) health disparity model "reads more like results.

Recommend rephrasing "Used thematic analysis to identify experiences at patient, primary care and specialist levels. Contributing factors identified using the National Institute of Minority Health and Health Disparities (NIHMD) health disparity model."

Response: Thank you for the recommendation. We have rephrased the methods to reflect the basis of the analysis as the NIMHD model alongside emerging themes related to suboptimal experiences.

3. Is the study design appropriate to answer the research question?

In a phenomenological study design the researchers is asked "to set aside their prejudices and a priori assumptions and focus mainly on the immediate experience." The first aim of the study was to "explore the pitfalls, suboptimal experiences and discriminatory practices in diagnostic pathways for BAA's with lung cancer." The authors assume the participants experienced pit falls, suboptimal experiences and discriminatory practices. This seems to be biased and not in concordance with phenomenological study design practice. A more appropriate aim may have been to "explore the lived experience of BAA's with lung cancer." Please clarify.

Response: We appreciate the perspective from which the reviewer approaches their critique. In a phenomenological exploration, it's advisable to embark with curiosity and openness. However, in a context like ours, the literature provides extensive evidence of pitfalls, suboptimal experiences, and prejudices. Arguably, if we are aware of these existing challenges, it could inadvertently bias the phenomenological experience if we don't

intentionally create space for these issues to surface. We did, in fact, set our intent clearly to identify and enumerate these challenges for better recognition. Our objective was not to portray the day-to-day life from waking to sleeping, but we did openly explore the diagnosis experience. We direct the reviewer to our interview guide for further reference. It features open-ended questions about the diagnostic experience from the perspectives of patients, primary care providers, and specialists. Only towards the end of each patient interview did we specifically ask about suboptimal experiences or invite reflections on perceived shortcomings.

Are the methods described sufficiently to allow the study to be repeated?

Yes

In the methods section – please delete the sentence “We identified suboptimal experiences during the diagnostic phase.” It belongs in the results.

Response: Thank you for the recommendation. We have reworded the sentence to capture the methodology on how we explored experiences, ultimately capturing the suboptimal

Page 6 line 54 please delete suboptimal as a descriptor for experiences. The descriptor assumes all experiences will be suboptimal. It would be helpful for the authors to describe the methods in objective terms.

Response: Thank you for the recommendation. We have reworded the descriptor of the experiences to be more capture the process leading up to identifying the sub optimal experiences, which did not happen in isolation.

Page 7 line 25 – please reword “patient who succumbed to advanced lung cancer.” Infers that it was the patient’s fault that he/she died. Suggest rewording to patient who died of advanced lung cancer – same comment for line 33.

Response: We have reworded the statement to patient who died of advanced lung cancer.

4. Are research ethics (e.g., participant consent, ethics approval) addressed appropriately?

Yes

5. Are the outcomes clearly defined?

No – the outcomes do not describe how many study participants described experiences for each of the pathways.

Response: Thank you for this observation. We were focussing on the depth of experiences of the participants. This was in the context of standard practices and likely to be reflective of other participant experiences.

6. If statistics are used, are they appropriate and described fully?

Not applicable

7. Are the references up-to-date and appropriate?

Suggest finding more recent publication.

Bach PB, Cramer LD, Warren JL, Begg CB. Racial differences in the treatment of early-stage lung cancer. *New England Journal of Medicine*. 1999;341(16):1198-205

Response: Thank you for your observation. We removed this citation and we included more recent ones.

Do the results address the research question or objective?

Yes

8. Are they presented clearly?

Yes

9. Are the discussion and conclusions justified by the results?

No – the statement “Our findings indicate that the majority of patients experienced delays in both the diagnosis and initiation of treatment” is not supported as the authors don’t provide any indication of the number of participants that experienced any of suboptimal experiences or factors in the study design. There are 1 or 2 supportive quotes for each suboptimal experience category included and no quantitative information on how many of the participants had a suboptimal experience in each of the suboptimal experience categories.

It would be helpful if the authors provided a count of how many patients/caregivers made comments about each of the suboptimal experiences along the diagnostic pathway and for each of the factors. With only one or two comments per experience and one per factor it’s hard to assess how prevalent these issues were in the study.

Response: Thank you for your thoughtful observation. We acknowledge that making claims about majority or minority without numerical support is not a standard practice in qualitative research. Given our small patient sample size (N<10 patients), we were concerned that presenting numbers might mislead the reader due to the inherently selective nature of the sample. This numerical assessment was outside the scope of this paper. To mitigate confusion, we revised our language to use "at least some" and “some” instead of "majority" and “many” indicating an occurrence without implying magnitude. It's important to note that our study's strength lies in its inclusion of diverse stakeholder perspectives, not only patients'. These viewpoints collectively facilitated our theme development. Nonetheless, we concur that it's essential to qualify such statements for accuracy.

10. Are the study limitations discussed adequately?

No

Included are small sample and highly educated and insured participants.

Not included – potential for bias due to study aim assumptions that participants have had suboptimal experiences. See previous comments.

Response: Thank you for your thoughtful critique. We understand your concern regarding the potential bias introduced by our study's focus on pitfalls, suboptimal experiences, and instances of discrimination in the diagnostic pathways for patients with lung cancer. We assure you that our study began with a broad exploration of all experiences before focusing on identifying specific challenges. This approach was taken to minimize any potential bias. We value your critique and will continue refining our methodology for future research. We added to the limitation section, *“Moreover, while our aim was to identify pitfalls, suboptimal experiences, and instances of discrimination, our focus on these specific areas might have inadvertently steered some participants to primarily discuss these aspects. It should be underscored, however, that we diligently pursued a broad exploration of all experiences initially, before directing our line of inquiry towards specific challenges.”*

11. Is the supplementary reporting complete (e.g., trial registration; funding details; CONSORT, STROBE or PRISMA checklist)?

Yes

12. To the best of your knowledge is the paper free from concerns over publication ethics (e.g., plagiarism, redundant publication, undeclared conflicts of interest)?

Yes

13. Is the standard of written English acceptable for publication?

Yes

* Statistical review

* 1. Does this paper require specialist statistical review?

No

Additional comments:

Introduction:

The sentence Whites, BAA patients over 65 years old with NSCLC had lower odds ratios of 0.53-0.63 for receiving targeted therapy “seems to be out of context here.

Response: We have reviewed and reworded the sentence to highlight the relationship between diagnosis and targeted therapy.

Please include references for the statement “Extensive literature exists on the roots of disparities and how to eliminate them for patients with early-stage lung cancers.”

Response: We included references 2, 29-31 for this statement.

Table 1 – participant characteristics – highly educated – 12 of 19 have doctorate, 2 Masters, 3 Bachelors, and 2 with some college – how generalizable are the results?

Response: Thank you for this observation. While this is a potential limitation which we have listed, it highlights the cross-cutting nature of disparities among BAAs irrespective of education status.

No discussion of smoking history – individuals who smoke or used to smoke are sometimes reluctant to seek care due to stigma of smoking and also more afraid of finding out they may have lung cancer. Suggest authors include smoking history in Table 1 and comment if participants who used to smoke or currently smoke expressed any self, implicit or explicit stigma related to their smoking history.

Response: Thank you for your insightful comment. We agree with your assertion that the stigma associated with smoking can sometimes lead to under-reporting due to an inherent desire to avoid potential blame. This, however, did not emerge as a theme in our specific data. While it is a pertinent topic, intentionally exploring the effects of smoking and stigma was not part of the scope of this study. We value your suggestion, and we will consider exploring smoking history into future investigations. We were very careful not to make people feel judged or blamed for their illnesses, an immediate reaction that comes among people who have lung cancer (the PI is a lung cancer survivor and co-authors are caregivers). We prioritized participants comfort and safety.

Page 13 – individual factors “Logistic constraints to accessing healthcare. Patients living in poorer areas tend to have poorer healthcare outcomes due to affordability and income issues. At times, patients live far from the health facility and are unable to access transportation or assistance to travel for treatment.” Most of the participants had incomes >100k – 9 of 19 >150, 3 100-150k – was it an issue for only a few participants?

Response: Thank you for this response. While some participants and caregivers had that experience, most of these factors were highlighted by oncologists, doctors and advocates who shared experiences working with their clients.

Page 13 line 52 “It has been observed that HCPs tend to neglect, miss, or dismiss symptoms reported by BAA patients, especially pain. Furthermore, they may not execute appropriate referrals” Is this from participant comments or from the literature? Please clarify.

Response: Thank you for this question. This was from literature, patient, and caregiver reports.

Page 15 line 6 – please use people first language – individual who smokes or smoked. Also is this point based on participant comments or on the literature.

Response: Thank you. We have amended the language to reflect person centred language. This point was raised by participants.

In the discussion of factors are these based on the literature or on lived experience from the healthcare professionals. Given the high education and socioeconomic status of the majority of the participants, many of these factors seem to not apply to this specific sample. It would be helpful to have demographic information on where the healthcare professionals practice (rural, urban, FQHC,

academic medical center...) and more descriptive demographics on where the patients and caregiver participants live i.e., urban, rural, underserved not just zip code.

Response: Thank you for this observation. Participants were drawn from across the Country and represent different demographic groups. While current information may provide insight, it may miss out on past experiences related to their personal, professional, and academic experiences beyond their current positions.

Page 31 – typo in second paragraph “We want you help in ...” should be “We want your help in ...”

Response: Thank you. We have made the correction.

There seems to be a typo on page 22 line 11 – the word “along” in the first sentence seems to be out of place.

Response: Thank you. We have made the correction and moved the word ‘along’.

Page 21 line 9 – please replace “hard to reach patients” with “underserved patients.” The term “hard-to-reach” implies patient is at fault for not being easy to reach.

Response: Thank you for the correction. We have amended the wording to be person centred.

Key Messages section:

The key message on how this study may affect research, policy and practice seems to be an overreach. It is unlikely that a small study with 19 highly educated participants is generalizable enough to lead to “positive changes in provider-patient interactions, identify areas for improvement in provider training and health systems, and improve cancer diagnosis and

treatment pathways for African Americans.” This study is more in the realm of hypothesis generation for future research that may lead to the improvements mentioned in the key message.

Response: We value the reviewer's perspective and acknowledge that a subsequent project exploring the prevalence of pitfalls, suboptimal experiences, and discriminatory practices, and quantitatively testing the hypotheses that the factors identified in this study are mechanistically implicated, would be a logical next step. We concur with the reviewer that our work, like much quality qualitative research, serves as the foundation for future studies.

Simultaneously, we emphasize the unique role of qualitative research in centering the voices of the community and amplifying the narratives of those who have lived the experiences. These aspects are crucial for advocacy and policymaking. Quantitative data and patient stories must work hand in hand to influence policy, improve training, and enhance the experiences of those impacted by cancer.

Reviewer: 2

Dr. Angela Giaquinto, American Cancer Society Comments to the Author:

This study provided qualitative interviews and evidence for adverse experiences among African American/Black people during the cancer care continuum specific to lung cancer. Overall, the study is well designed with clear and concise explanation of methods used. This reviewer suggests some additional information to be added to the results if collected to provide further evidence for statements given (see results section below). This reviewer also encourages the authors to review their

manuscript for consistency in capitalizations. For instance, the introduction had variations in capitalizing “white”, Table 1 has inconsistent capitalizations of gender, race/ethnicity, employment, and role, etc. Table 2 had supportive quotes capitalized in the second and third column but not the first, etc. Below are additional comments for the authors’ consideration.

Response: Thank you for your observation. We have edited the sections to be consistent in the sections highlighted and corrected any typos.

Introduction: The introduction included a good background. Page 5 line 7-8, I encourage the researchers to use the most recent 5-year data (2016-2020) available from NCHS for mortality or list the years that the given statistic is for as 55.4 and 46.9 per 100,000 for lung cancer death rates are not from 2016-2020.

Response: We removed the numbers taken from this reference and included only the more recent numbers. Thank you for the observation.

Page 5 line 27-30 has a sentence repeated, the first time has reference 16 and the duplicate sentence has references 17 and 18. Please double check references as I believe 16 should be referenced in the prior sentence beginning on line 24.

Response: Thank you for the observation. We have corrected the section by updating the reference and deleted the repeated sentence.

Page 6 first paragraph, I encourage the authors in line 4 to include “diagnostic pathways for BAAs with metastatic/late-stage lung cancer.” To tie in the last paragraph on page 5.

Methods: The methods were well written, clear, and concise.

Results: If possible, this reviewer suggests that the authors include a table that gives qualitative evidence for claims, i.e., how many patients interviewed expressed similar individual factors that led to suboptimal experiences.

Response: Thank you for the recommendation. The goal of this paper is to outline various experiences, rather than to quantify the prevalence of each. As you know, the study was not designed with explicit inquiries about each issue or factor. Consequently, while some participants spontaneously raised concerns or shared about a certain factor, others might have omitted sharing about a similar experience or relevant factor simply because it was not at the forefront of their minds. We are concerned that providing a table may not adequately represent the issue at hand. Quantifying the problem and further assessing the relevance of these factors are tasks left to future researchers. We acknowledge the importance of this area and currently mention it in both the limitations of our work and as a topic for future research.

Discussion states on page 19 line 18 that a “majority of patients experienced delays in diagnosis and initiation of treatment”. Can this data be included in the results? I.e., including a table or summarized findings from Question b viii.

Response: Thank you for your question. Based on your comment, and the comment of Reviewer 1, we have revised our original statement to indicate occurrence, but have refrained from providing a quantitative assessment. As you know, our study, with its smaller sample size, did not aim to select a sample of participants to be random or necessarily representative of all range of experiences. Consequently, it would be inappropriate for us to comment on the magnitude of the problem. However, we can assert that a certain experience occurred among some participants, as currently stated. This delay is explicitly mentioned in the Results section under the PCP/Hospital/ER/Urgent Care phase, as evidenced by the following phrase: "Delayed or not offered diagnostic imaging. Some patients were not offered diagnostic tests due to the healthcare professional’s low levels of suspicion for lung cancer." This delay is also

implied during the Secondary Care phase, with mentions of difficulties in securing appointments to receive care.

Discussion: Well written. This reviewer was surprised that 14 of the 19 interviewees had post-graduate education and only 1 patient reported an income less than 50,000-100,000. This highly educated and compensated population may not be generalizable, which the authors including in their limitations. This reviewer agrees that they study’s strength was the in-depth interviews and authentic perspectives.

Response: Thank you for highlighting this. Future research will help unveil to what extent these factors affect other populations.

Overall, this is a well-written article that has a clearly defined study objective. I encourage the authors to include, if possible, more information on aggregated responses collected to further strengthen their discussion, i.e., how many patients had similar responses or experienced delays in treatment. This reviewer recommends the paper be accepted with minor revisions.

Response: We thank you for your review. We will include more quantitative data in our next study to address and highlight the issues as recommended. We addressed this in the limitation and future direction.

VERSION 2 – REVIEW

REVIEWER	Borondy Kitts, Andrea Rescue Lung Society
REVIEW RETURNED	04-Sep-2023

GENERAL COMMENTS	Thank you to the authors for addressing the comments from my first review. This is a well written paper exploring types of suboptimal experiences encountered by Blacks in diagnosis of lung cancer. The use of the NIMHD model as a framework for the contribution factors is helpful in identifying potential causes and interventions. I have just a couple of minor suggested revisions. 1. Using the word remarkable in the abstract conclusion is confusing and may be a biased observation. Suggest removing and sticking to the facts - for example “BAA patients and caregivers encountered suboptimal experiences during their care. The NIMHD model is a useful framework to organize factors contributing to these experiences that may be leading to health disparities.” 2. In the response to reviewers letter the authors state “This project aims to map the range of suboptimal experiences, which can then be included in future surveys to screen patients for these occurrences.” I find that the authors have not clearly stated this in their manuscript. Informing the development of a screening tool for suboptimal experiences would be a major contribution to identifying the extent of suboptimal experiences. I suggest the authors include in the discussion that their study provides information that can help inform the development of screening surveys for BAA patients and caregivers to ascertain the
---

	existence, extent, and nature of suboptimal experiences. Identifying suboptimal experiences provides opportunity to develop and implement interventions to mitigate harms in those who have suboptimal experiences and optimally to prevent them from happening.
REVIEWER	Giaquinto, Angela American Cancer Society
REVIEW RETURNED	06-Sep-2023
GENERAL COMMENTS	The author adequately addressed major comments raised by this author in their revision of the manuscript.

VERSION 2 – AUTHOR RESPONSE

Reviewer: 1

Dr. Andrea Borondy Kitts, Rescue Lung Society

Comments to the Author:

Thank you to the authors for addressing the comments from my first review. This is a well written paper exploring types of suboptimal experiences encountered by Blacks in diagnosis of lung cancer. The use of the NIMHD model as a framework for the contribution factors is helpful in identifying potential causes and interventions.

I have just a couple of minor suggested revisions.

1. Using the word remarkable in the abstract conclusion is confusing and may be a biased observation. Suggest removing and sticking to the facts - for example “BAA patients and caregivers encountered suboptimal experiences during their care. The NIMHD model is a useful framework to organize factors contributing to these experiences that may be leading to health disparities.”

Response: Thank you for suggesting this revision. We used the statement you suggested.

2. In the response to reviewers letter the authors state “This project aims to map the range of suboptimal experiences, which can then be included in future surveys to screen patients for these occurrences.” I find that the authors have not clearly stated this in their manuscript. Informing the development of a screening tool for suboptimal experiences would be a major contribution to identifying the extent of suboptimal experiences. I suggest the authors include in the discussion that their study provides information that can help inform the development of screening surveys for BAA patients and caregivers to ascertain the existence, extent and nature of suboptimal experiences. Identifying suboptimal experiences provides opportunity to develop and implement interventions to mitigate harms in those who have suboptimal experiences and optimally to prevent them from happening.

Response. We added this statement to the discussion. “This project helped mapping the range of suboptimal experiences, which can then be included in future surveys to screen patients for these occurrences. Future research should aim...”